# Hilbert Space Delocalization under Random Unitary Circuits

**DOI:** 10.3390/e26060471

**Published:** 2024-05-29

**Authors:** Xhek Turkeshi, Piotr Sierant

**Affiliations:** 1Institut für Theoretische Physik, Universität zu Köln, Zülpicher Strasse 77a, 50937 Cologne, Germany; 2ICFO—Institut de Ciències Fotòniques, The Barcelona Institute of Science and Technology, Av. Carl Friedrich Gauss 3, 08860 Castelldefels, Barcelona, Spain; psierant@icfo.net

**Keywords:** quantum circuits, many-body quantum dynamics, quantum coherence

## Abstract

The unitary dynamics of a quantum system initialized in a selected basis state yield, generically, a state that is a superposition of all the basis states. This process, associated with the quantum information scrambling and intimately tied to the resource theory of coherence, may be viewed as a gradual delocalization of the system’s state in the Hilbert space. This work analyzes the Hilbert space delocalization under the dynamics of random quantum circuits, which serve as a minimal model of the chaotic dynamics of quantum many-body systems. We employ analytical methods based on the replica trick and Weingarten calculus to investigate the time evolution of the participation entropies which quantify the Hilbert space delocalization. We demonstrate that the participation entropies approach, up to a fixed accuracy, their long-time saturation value in times that scale logarithmically with the system size. Exact numerical simulations and tensor network techniques corroborate our findings.

## 1. Introduction

Starting from a vector of a generically chosen basis of the Hilbert space, unitary quantum dynamics generate a superposition spanning the entire basis of the Hilbert space. This phenomenon, referred to as Hilbert space delocalization, can be viewed as the spreading of the many-body wave function over the Hilbert space under quantum dynamics. Hilbert space delocalization is tied to non-equilibrium processes in quantum mechanics and, hence, pivotal in our understanding of quantum foundations [1], quantum technologies [2,3], and condensed matter theory [4,5,6]. Being intimately connected to the resource theories of quantum coherence and entanglement [7,8], Hilbert space delocalization provides a valuable tool for the characterization of quantum phases of matter [9,10,11,12,13,14], quantum chaos and thermalization [15,16,17,18,19,20,21,22], and their violations in non-ergodic systems—including monitored systems [23,24,25,26,27,28] and disorder-induced localization [29,30,31,32,33,34,35,36,37,38,39,40,41,42,43,44,45,46].

Natural quantifiers for the localization and delocalization properties of a state ρ=|Ψ〉〈Ψ| over the many-body Hilbert space are the inverse participation ratios (IPRs) and the participation entropies, which measure the spreading of the state distribution $p_{n}\equiv \langle n|\rho |n\rangle$ over the basis B={|n〉}. The IPR and participation entropy are given, respectively, by
(1)$$I_{q}\equiv \sum_{n\in B}{\left(\langle n|\rho |n\rangle \right)}^{q}=\sum_{n\in B}p_{n}^{q}\;,\;S_{q}=\frac{1}{1-q}\ln\left[I_{q}\right]\;.$$
(We note the connection with the relative entropies of coherence [47] $C_{q}\left(\rho ,B\right)\equiv S_{q}-{\left(1-q\right)}^{-1}\ln\mathrm{tr}\left(\rho^{q}\right)$, a scalable quantifier in quantum coherence resource theory. In particular, for pure states, these two quantities coincide, $C_{q}\left(\rho ,B\right)=S_{q}$).

Despite the explicit basis dependence, extensive studies have demonstrated that the IPR captures the structural properties of quantum matter, including universal behavior at phase transitions [26,48]. However, the investigations of the IPR and participation entropies have focused so far on the equilibrium and stationary state features, leaving the question of the time evolution of quantum coherence, and, hence, of Hilbert space delocalization, unresolved.

In this work, we investigate Hilbert space delocalization under the dynamics of (1+1)D quantum circuits comprising 2-qudit Haar random gates arranged in a brick wall pattern [23]. The locality and unitarity of this setup constitute the minimal requirements for chaotic evolution in many-body systems [49,50,51,52,53,54,55,56,57,58,59,60,61,62,63,64,65,66,67,68,69,70], hence allowing us to gain a phenomenological understanding of the Hilbert space delocalization under generic quantum dynamics. Our analysis combines rigorous analytical methods, based on the mapping between the average of the IPR over the random circuits and a statistical mechanics model, with exact numerical simulations, including tensor network techniques [71,72,73,74] (here, our implementation is based on the open-source library ITensor [75,76] and available at [77]). We find that the saturation of participation entropy $S_{q}$ to the long-time stationary value, equal to the random Haar state average ${\tilde{S}}_{q}^{H}$, occurs exponentially quickly in time (i.e., in circuit depth):(2)$$S_{q}={\tilde{S}}_{q}^{H}-\alpha_{q}N{\left(2K_{d}\right)}^{t-1},$$
where $K_{d}>0$ is a constant characterizing the properties of 2-qudit gates, and $\alpha_{q}>0$ is a constant. We find an exact expression for the constant $K_{d}$ for Haar random 2-qudit gates and provide analytic arguments for the value of $\alpha_{q}$ in the limit of large on-site Hilbert space dimension. In other words, Equation (Equation 2) implies that the participation entropy $S_{q}$ approaches its stationary value, up to a fixed accuracy, at a time scaling logarithmically with the system size, $\tau_{\mathrm{HSD}}∼\ln\left(N\right)$.

The paper is structured as follows. In Section 2, we outline the toolbox employed in the rest of the paper, and, in Section 3, we discuss the stationary value of participation entropy that the deep quantum circuit reaches. Section 4 presents the core of our analytical approach based on the mapping of the IPR calculation to a statistical mechanics problem. We resolve the Rényi-2 participation entropy evolution in Section 5, where we present the analytic prediction. These findings are further corroborated by the numerical analysis in Section 6 for generic Rényi index *q*. We present the outlook of our work and a discussion of its further implications in Section 7. The more technical parts of the paper are detailed in the appendix. In Appendix A, we present a self-contained discussion about Haar averages. In Appendix B, we review the interface problem related to the entanglement propagation and, in Appendix C, briefly discuss the numerical implementations.

## 2. Methods

We begin by presenting the analytical toolbox we will employ in the rest of the paper. The latter is based on the replica trick, the superoperator formalism [78], and Weingarten calculus [79,80,81,82] (see also Refs. [54,61]). Since 〈n|•|m〉=tr(|m〉〈n|(•)) and $\mathrm{tr}{\left(AB\right)}^{q}=\mathrm{tr}\left(A^{⊗q}B^{⊗q}\right)$, the IPR definition (Equation 1) implies that $I_{q}=\mathrm{tr}\left(\Lambda_{B}\rho^{⊗q}\right)$, where $\Lambda_{B}=\sum_{n\in B}{\left|n\rangle \langle n\right|}^{⊗q}$ is a replica operator acting on $H_{N}^{⊗q}$. Throughout this manuscript, we will study qudits with local Hilbert space dimension *d*; hence, $H_{N}≃C^{d^{N}}$. For concreteness, we will fix B to be the computational basis, where $\left|n\rangle =\right|b^{1},\ldots ,b^{N}\rangle$ with $b_{j}=0,\ldots ,d-1$ for each *j*. Nevertheless, due to the invariance of the considered circuits under local basis rotations, our results remain valid for any basis of $H_{N}$ obtained by a unitary transformation $U=U_{1}⊗\ldots ⊗U_{N}$ of the computational basis, where $U_{k}$ belongs to the unitary group U(d) for a single qudit. For the computational basis, $\Lambda_{B}={⨂}_{j=1}^{N}\Lambda_{q}^{\left(j\right)}$, which defines the “book” boundary condition for each qudit [83,84]:(3)$$\Lambda_{q}^{\left(j\right)}\equiv \sum_{b^{j}=0,\ldots ,d-1}(|b^{j}\rangle {\langle b^{j}|)}^{⊗q}\;,$$
where the superscript in $\Lambda_{q}^{\left(j\right)}$ bookkeeps the number of the affected qudit. For later convenience, we recast the problem in the superoperator formalism. For any operator *U* and *A*, we have $A=\sum_{n,m=0}^{d^{N}-1}A_{n,m}|n\rangle \langle m|\mapsto |A⟫=\sum_{n,m=0}^{d^{N}-1}A_{n,m}|n,m⟫$, and $UAU^{†}\mapsto \left(U⊗U^{*}\right)|A⟫$ (we denote with ${\left(•\right)}^{*}$ and ${\left(•\right)}^{†}$, respectively, the complex and Hermitian conjugation of •). In this representation, $⟪B|A⟫=\mathrm{tr}(B^{†}A)$, so the IPR can be written as
(4)$$I_{q}=⟪\Lambda_{q}^{\left(1\right)},\Lambda_{q}^{\left(2\right)},\ldots ,\Lambda_{q}^{\left(N\right)}|\rho^{⊗q}⟫.$$

For convenience, we introduce a graphical notation. We implicitly define it for the replica boundary $\Lambda_{q}$ and the replica state $\rho^{⊗q}$ via the IPR as follows: (5)
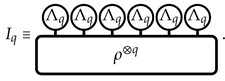


Lastly, since we will consider the unitary evolution $|\rho ⟫=\left(U⊗U^{*}\right)|\rho_{0}⟫$ for some $|\rho_{0}⟫$ values selected as the initial state and unitary operation *U* acting on the Hilbert space $H_{N}$, we introduce the following graphical notation for *U* and $U^{*}$, respectively: (6)
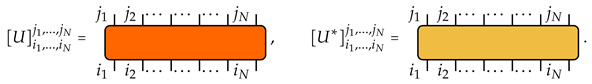


We will omit the legs indices, which can be inferred from the context. Moreover, we will typically reserve thick (thin) lines for multi-replica (single-replica) objects.

## 3. Delocalization Properties of Random Haar States

We begin our discussion by reviewing the Hilbert space delocalization of random states [21,83] which correspond to the stationary ensemble of states obtained under the action of sufficiently deep random circuits. Uniformly distributed random states in the Hilbert space $|\rho_{\mathrm{Haar}}⟫=\left(U⊗U^{*}\right)|\rho_{0}⟫$ are obtained from a reference state $|\rho_{0}⟫=|\Psi_{0},\Psi_{0}\rangle$ via a unitary operation *U* acting globally on the system of *N* qudits, drawn with the Haar measure from the *N* qudit unitary group $U(d^{N})$, where *d* is the qudit local Hilbert space dimension. Let us compute the average IPR over the Haar ensemble, which reads, using the graphical notation introduced in Section 2, as
(7)
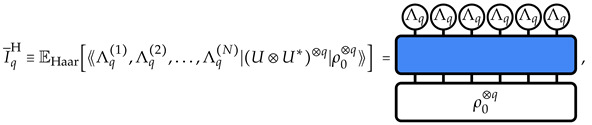

where $E_{\mathrm{Haar}}$ denotes the expectation value over the unitary group $U(d^{N})$ taken with the Haar measure. Here, we have defined, via the linearity of the average and of the expectation value, the *q*-replica transfer matrix on *N* qudits: (8)
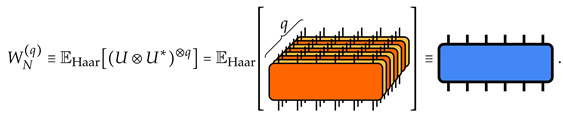


The explicit formula for $W_{N}^{\left(q\right)}$ can be obtained by a direct evaluation of the Haar average, as detailed in Appendix A. The final result is expressed in terms of the permutation operators ${|\sigma ⟫}_{j}$ acting on each qudit *j*, with the matrix elements in the superoperator computational basis given as
(9)$$⟪b_{1}^{j},{\bar{b}}_{1}^{j},b_{2}^{j},{\bar{b}}_{2}^{j},\ldots ,b_{q}^{j},{\bar{b}}_{q}^{j}|\sigma ⟫j=\prod_{k=1}^{q}\delta_{b_{k}^{j}\;,\;{\bar{b}}_{k}^{\sigma (j)}}\;,$$
fixed by the permutation $\sigma \in S_{q}$. We employ the Weingarten function Wg(D;σ) allows us to define the dual states $|{\tilde{\sigma }⟫}_{1,2,\ldots ,N}=\sum_{\tau \in S_{q}}\mathrm{Wg}\left(d^{N};\sigma \tau^{-1}\right){|\tau ⟫}_{1}⊗\cdots ⊗|\tau {⟫}_{N}$ acting on the whole system of *N* qudits. With these states, the transfer matrix reads
(10)
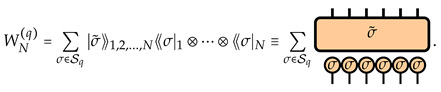


We are now in position to compute the IPR. Since the initial state $\rho_{0}$ is pure, $⟪\sigma ,\ldots ,\sigma |\rho_{0}^{⊗q}⟫=1$. Furthermore, using $\sum_{\sigma }\mathrm{Wg}\left(d^{N};\sigma \tau^{-1}\right)=\left(d^{N}-1\right)!/\left(d^{N}+q-1\right)$ [81], we have
(11)
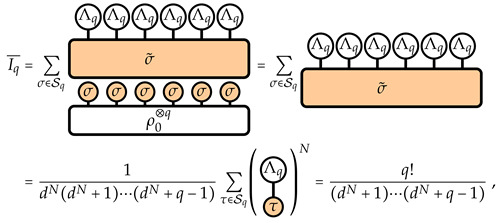

where we use $|S_{q}|=q!$ and $⟪\Lambda_{q}|\tau ⟫=d$ for all $\tau \in S_{q}$. From this calculation, it follows that the *annealed averaged* Rényi participation entropy for the Haar states is given by
(12)$${\tilde{S}}_{q}^{H}\equiv \frac{1}{1-q}\ln\left[{\bar{I}}_{q}^{H}\right]=\frac{\ln\left[\left(d^{N}+1\right)\ldots \left(d^{N}+q-1\right)\right]-\ln\left[q!\right]}{q-1}\overset{N\to \infty }{⟶}N\ln\left[d\right]-\frac{1}{q-1}\ln\left[q!\right]\;.$$

As expected and already discussed in [21], the Haar random states are (almost) fully delocalized over the many-body basis B. Indeed, (Equation 12) shows that ${\tilde{S}}_{q}^{H}$ differs only by a sub-leading constant term ${\left(q-1\right)}^{-1}\ln\left[q!\right]$ from the maximal value of participation entropy Nln(d) achieved for a uniformly distributed state with $\langle n|\rho |n\rangle =d^{-N}$. Since the logarithm is non-linear, in principle, one should expect corrections when considering the *quenched average* of the participation entropy ${\bar{S}}_{q}^{H}\equiv E_{\mathrm{Haar}}\left[S_{q}\right]$. We briefly estimate these fluctuations via the variance (for the explicit expression of $\bar{I_{q}^{r}}$, see Ref. [83]).
(13)$$\mathrm{std}\left(I_{q}\right)\equiv \sqrt{\bar{I_{q}^{2}}-{\left(\bar{I_{q}}\right)}^{2}}\;.$$

For this expression, we need to compute $\bar{I_{q}^{2}}$, which requires the use of 2q replicas of $H_{N}$ and the boundary conditions $⟪\Lambda_{q}⊗\Lambda_{q}|$ acting on each site. Algebraic manipulations analogous to the ones described above give
(14)
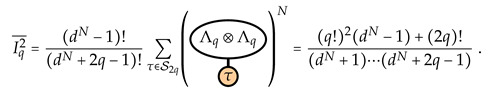


It follows that $\mathrm{std}\left(I_{q}\right)∼O\left(d^{(1-2q)N/2}\right)$, and, in the thermodynamic limit, these fluctuations are irrelevant. In particular, in the scaling limit N→∞, the annealed and the quenched averages coincide, ${\tilde{S}}_{q}^{H}-{\bar{S}}_{q}^{H}\overset{N\to \infty }{⟶}0$, and, with probability equal to unity, approximate the value of the participation entropy calculated for a single Haar random state with an arbitrarily small fixed accuracy.

## 4. Hilbert Space Delocalization in Brick Wall Quantum Circuits

After this preliminary discussion, relevant to the deep circuit limit, we now discuss the Hilbert space delocalization under random quantum circuits. As the initial state, we fix $|\Psi_{0}{\rangle =|0\rangle }^{⊗N}$ and study the approach of the annealed and quenched average participation entropy to the asymptotic value ${\tilde{S}}_{q}^{H}={\left(1-q\right)}^{-1}\ln\left[I_{q}^{H}\right]$ with $I_{q}^{H}\equiv q!\left(d^{N}\right)!/\left(d^{N}+q-1\right)!$, cf. Equations (Equation 11) and (Equation 12). (For the circuit case, the quenched and annealed averages are given, respectively, by
(15)$${\bar{S}}_{q}=E_{\mathrm{Haar}}\left[{\left(1-q\right)}^{-1}\ln I_{q}\left(t\right)\right]\;,\;{\tilde{S}}_{q}={\left(1-q\right)}^{-1}\ln E_{\mathrm{Haar}}\left[I_{q}\left(t\right)\right]\;,$$
with $I_{q}\left(t\right)=\sum_{b}\left|\langle b\right|U_{t}|\Psi_{0}{\rangle |}^{2q}$). The evolution operator corresponding to the brick wall quantum circuit of depth *t* (referred to also as time) is given by
(16)$$U_{t}=\prod_{\tau =0}^{t-1}U^{\left(\tau \right)}\;,\;U^{\left(\tau \right)}=\prod_{i=0}^{N/2-\left(\tau \%2\right)}U_{2i+\left(\tau \%2\right),2i+1+\left(\tau \%2\right)}\;,$$
where m%k denotes the integer *m* modulo and the integer *k*. Here, each $U_{m,n}$ is an independent identically distributed (i.i.d.) Haar random unitary from $U(d^{2})$ acting on the *m*-th and *n*-th qudit. Our goal is to compute the
(17)$${\bar{I}}_{q}\equiv E_{\mathrm{Haar}}\left[⟪\Lambda_{q},\ldots ,\Lambda_{q}\left|{\left(U_{t}⊗U_{t}^{*}\right)}^{⊗q}\right|\psi_{0},\psi_{0},\ldots ,\psi_{0}⟫\right]\;,$$
where the *N* copies of $|\psi_{0}⟫={|0\rangle }^{⊗2q}$ implement the initial condition $|\Psi_{0}\rangle$ on the replica space. We note that the $E_{\mathrm{Haar}}$ denotes here the average over the realizations of the two-body gates $U_{m,n}$ which comprise the considered brick wall circuit. We adapt the graphical notation of Section 2 for a generic two-body gate *U* with indices i,j,k,l∈{0,d−1} numbering the states of the qudits: (18)
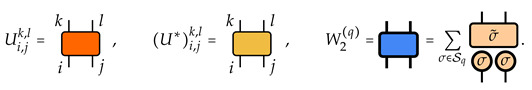


Recalling that the two-body gates $U_{m,n}$ are i.i.d. variables, we observe that their averages factorize, implying that the transfer matrix corresponding to the circuit (Equation 16) of depth *t* reads
(19)
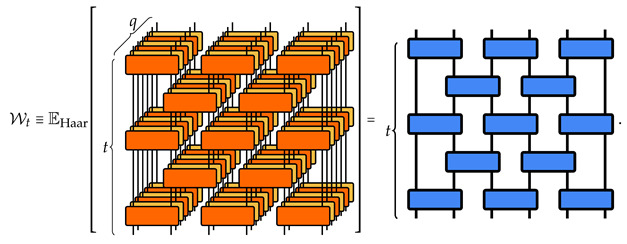


Contracting with the replica boundaries, and recalling the decomposition for $W_{2}^{\left(q\right)}$ in Equation (Equation 18), we obtain the final expression
(20)
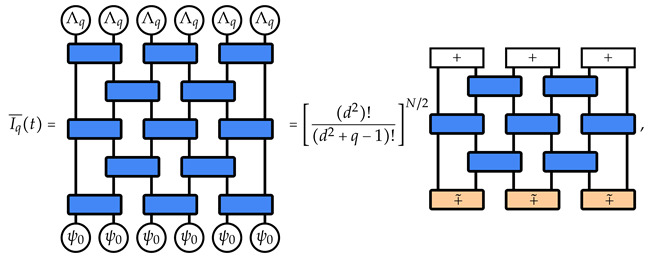

where, denoting $|{\tilde{\sigma }⟫}_{i,i+1}=\sum_{\tau \in S_{q}}\mathrm{Wg}\left(d^{2};\sigma \tau^{-1}\right){|\tau ⟫}_{i}⊗{|\tau ⟫}_{i+1}\;$, we use $⟪\sigma |\psi_{0}⟫=1$ for each site, $(⟪\Lambda_{q}{|}_{i}⊗⟪\Lambda_{q}{|}_{i+1})|{\tilde{\sigma }⟫}_{i,i+1}=d^{2}$, and we define ${|+⟫}_{i,i+1}=\sum_{\sigma \in S_{q}}{|\sigma ⟫}_{i}⊗{|\sigma ⟫}_{i+1}$ and ${|\tilde{+}⟫}_{i,i+1}=\sum_{\sigma \in S_{q}}{|\tilde{\sigma }⟫}_{i,i+1}$. (We note that the contraction between the $W_{2}^{\left(q\right)}$ requires ${⟪{\tau |}_{i}\cdot |\sigma ⟫}_{i}=d^{\#(\sigma \tau^{-1})}$, with #(σ) the number of cycles of the permutation σ. These weights can be reabsorbed in $W_{2}^{\left(q\right)}$ via a local basis change, see Appendix A).

Equation (Equation 20) is one of the main results of this work as it already allows for efficient tensor network implementations [75]. At the same time, Equation (Equation 20) can be viewed as the partition function of a statistical mechanics model once the $W_{2}^{\left(q\right)}$ values are replaced with decomposition Equation (Equation 18). To highlight this interpretation, we can unravel the upper boundary condition as
(21)$$|Q_{\mathrm{in}}⟫=\sum_{\left\{\sigma_{j}\in S_{q}\right\}}|Q_{\mathrm{in}}\left(\left\{\sigma_{j}\right\}\right)⟫\;\;|Q_{\mathrm{in}}\left(\left\{\sigma_{j}\right\}\right)⟫\equiv \overset{N/2}{\underset{i=1}{⨂}}{|\sigma_{j}⟫}_{i}⊗{|\sigma_{j}⟫}_{i+1}\;.$$

It follows that $W_{t}$ is a sum of backward paths from each initial state $|Q_{\mathrm{in}}(\left\{\sigma_{j}\right\})⟫$ to each final $|Q_{\mathrm{fin}}(\left\{\sigma_{j}\right\})⟫\equiv {⨂}_{i=1}^{N/2}{|{\tilde{\sigma }}_{j}⟫}_{i}⊗{|\sigma_{j}⟫}_{i+1}$.

We conclude this section by highlighting the similarities and the differences between the IRP calculation and the computations for propagation of entanglement entropy [54,55], defined as $S_{q}^{\mathrm{ent}}=-\ln\left[\mathrm{tr}\left(\rho_{A}^{q}\right)\right]$, where $\rho_{A}={\mathrm{tr}}_{A_{c}}\left(\rho \right)$ is the reduced density matrix of the pure state ρ=|Ψ〉〈Ψ| and $A\cup A_{c}$ a bipartition of the system. As for the participation entropy, we focus on the annealed averages, determined by the purities, i.e., the reduced density matrix moments $P_{q}\equiv E_{\mathrm{Haar}}\left[\mathrm{tr}\left(\rho_{A}^{q}\right)\right]$. Denoting the identity permutation as $I\in S_{q}$() and the cyclic permutation as $S\in S_{q}$(12…q), in the superoperator formalism, we have
(22)$$P_{q}\left(t;A\right)=⟪Q_{\mathrm{in}}\left(\left\{S_{j}:j\in A,I_{l}:l\in A_{c}\right\}\right)\left|W_{t}^{\left(q\right)}\right|\psi_{0}^{⊗N}⟫\;,$$
where $|Q_{\mathrm{in}}(\left\{S_{j}:j\in A,I_{l}:l\in A_{c}\right\})⟫$ is a product of identity (cyclic) operators on the qudit *j*, for $j\in A_{c}$ (j∈A). The statistical mechanics model corresponding to the calculation of $P_{q}\left(t;A\right)$ is exactly the same as Equation (Equation 20) but with different boundary conditions; for the IPR, we have free boundaries, cf. Equation (Equation 21), while, for the $P_{q}\left(t;A\right)$, the boundary conditions are of the domain wall type. This fact leads to substantial differences between the Hilbert space delocalization and the entanglement propagation in the considered random circuits, which we will discuss in the next section focusing on the two-replica limit. We will further corroborate these findings with a numerical analysis for various replica numbers (Rényi index) *q*.

## 5. Two-Replica Computations

In the two-replica limit, the parallelism between the inverse participation ratios and the reduced density matrix moments is more direct. Since $S_{2}=\left\{I,S\right\}$, Equation (Equation 21) translates to the sum over all domain wall boundary conditions (Equation 22). Therefore, the IPR, up to the overall constant, is given by the sum of $P_{2}$ over all possible choices of subsets *A* of contiguous pairs of sites Q≡{{1,2},{3,4},…,{N−1,N}}, cf. Equations (Equation 20) and (Equation 21), resulting in
(23)$${\bar{I}}_{2}\left(t+1\right)=\frac{1}{{\left(d^{2}+1\right)}^{N/2}}\sum_{A\subset Q}P_{2}\left(t;A\right)\;,$$
where the shift to time t+1 comes from the simplification of one layer in Equation (Equation 20). The time evolution of the purities [54,55] can be split into two terms:(24)$$P_{2}\left(t;A\right)=\frac{d^{2N_{A}}+d^{2(N/2-N_{A})}}{d^{N}+1}+f_{A}\left(t\right)\;,$$
where the first term describes the long-time saturation value of the purity, the second term parametrizes the approach to the saturation value, and the factor 2 in the exponents arises because the set Q contains N/2 pairs of neighboring lattice sites. Combining Equations (Equation 23) and (Equation 24) leads to
(25)$${\bar{I}}_{2}\left(t+1\right)=\frac{1}{{\left(d^{2}+1\right)}^{N/2}}\sum_{A\subset Q}\left(\frac{d^{2N_{A}}+d^{2(N/2-N_{A})}}{d^{N}+1}+f_{A}\left(t\right)\right)=I_{2}^{H}+\frac{1}{{\left(d^{2}+1\right)}^{N/2}}\sum_{A\subset Q}f_{A}\left(t\right)\;,$$
where the first term in Equation (Equation 24) is independent of the partition. This fact implies that
∑A⊂Qd2NA+d2(N/2−NA)=2∑NA=0N/2N/2NAd2NA=2(d2+1)N/2,
and, as a result, we have
(26)$${\tilde{S}}_{2}^{H}-{\tilde{S}}_{2}\left(t\right)=\ln\left[1+\frac{d^{N}+1}{2{\left(d^{2}+1\right)}^{N/2}}\sum_{A\subset Q}f_{A}\left(t-1\right)\right].$$

We are interested in identifying the leading terms governing the decay of ${\tilde{S}}_{2}^{H}-{\tilde{S}}_{2}\left(t\right)$ at long times. Fixing the system size *N* and considering the limit of d≫N, the factor $\left(d^{N}+1\right)/\left[2{\left(d^{2}+1\right)}^{N/2}\right]$ in Equation (Equation 26) simplifies to unity (we note that the d→∞ limit has been extensively considered in the literature, see Ref. [49]). We focus first on the single domain wall configurations, corresponding to bipartitions of the system into $A=\{1,2,\ldots ,2N_{A}\}$ and $A_{c}=\left\{2N_{A}+1,\ldots ,N\right\}$. The exact solution for the single domain wall case, detailed in Appendix B, shows that $f_{A}\left(t\right)\propto {\left(2K_{d}\right)}^{t}$ for domains localized in the bulk of the system, where
(27)$$K_{d}=\frac{d}{d^{2}+1}.\;$$

The time evolution of the purities $P_{2}\left(t;A\right)$, where *A* corresponds to a configuration with $n_{d}$ domains, is suppressed by the factor ${\left(2K_{d}\right)}^{n_{d}t}$, which vanishes at any t>0 in the large *d* limit. Therefore, the leading contributions to the ${\tilde{S}}_{2}^{H}-{\tilde{S}}_{2}\left(t\right)$ arise due to the single domain configurations. The number of the relevant single domain wall configurations scales proportionally to the system size *N*, translating Equation (Equation 26) to
(28)$${\tilde{S}}_{2}^{H}-{\tilde{S}}_{2}\left(t\right)\overset{d\gg 1}{⟶}\alpha_{d}N{\left(2K_{d}\right)}^{t-1}$$
where the coefficient $\alpha_{d}\overset{d\gg 1}{⟶}1/2$ at sufficiently large system size *N*. While we have used the large on-site Hilbert space dimension limit to derive Equation (Equation 28), we anticipate that this equation applies to any d≥2 with a properly chosen factor $\alpha_{d}$.

We are unable to analytically demonstrate the validity of Equation (Equation 28) at a fixed finite value of *d* and for N≫1. Nevertheless, we outline the underlying calculations, which allow us to pin-point the leading factors in the time dependence of ${\tilde{S}}_{2}^{H}-{\tilde{S}}_{2}\left(t\right)$. First, we observe that, at fixed *d* and for N≫1, the contributions from the exponentially many configurations A⊂Q are exponentially in *N* suppressed by the term $\left(d^{N}+1\right)/\left[2{\left(d^{2}+1\right)}^{N/2}\right]$. Performing tensor network contractions, we find for a generic initial configuration that $f_{A}\left(t\right)=a_{A}{\left(2K_{d}\right)}^{t}$ in the long-time limit. Interpreting the tensor network contraction as the wandering domain walls problem (cf. Appendix B), the factor ${\left(2K_{d}\right)}^{t}$ corresponds to the single domain wall configuration which dominates $f_{A}\left(t\right)$ at long times, while the factor $a_{A}$ depends on the processes required to reach the single domain wall configuration from the initial condition. The summation over all initial conditions yields an overall factor ${\left(2K_{d}\right)}^{t}$ which describes the behavior of ${\tilde{S}}_{2}^{H}-{\tilde{S}}_{2}\left(t\right)$ at any d≥2. We conjecture that the competition of the renormalizing factors $a_{A}$ with the exponential terms appearing in the summation over A⊂Q yields a factor proportional to the system size *N* at any d≥2. This conjecture is equivalent to the validity of Equation (Equation 28) at any d≥2.

To corroborate our analytical considerations, we compare our predictions with the numerical simulations of the brick wall random circuits varying the on-site Hilbert space dimension *d*. A brief summary of our numerical implementations is detailed in Appendix C. First, we numerically calculate the exact time evolution of the system’s state |Ψ〉 up to the system size N=24. Our findings are reported in Figure 1 (left). We observe that, already at the time scale O(1), the difference between the annealed average ${\bar{S}}_{2}$ and the quenched average ${\tilde{S}}_{2}$ of the participation entropy is negligible. Thus, the self-averaging properties of the IPR and participation entropies derived in the long-time limit in Section 3 apply also for the relatively shallow circuits, allowing us to use the ${\tilde{S}}_{2}\left(t\right)=-\ln\left[{\bar{I}}_{2}\left(t\right)\right]$ as an accurate proxy for the circuit-averaged participation entropy ${\bar{S}}_{2}\left(t\right)$.

To test our analytical prediction (Equation 28), we focus on the difference between the stationary Haar value ${\tilde{S}}_{2}^{H}$ and the annealed average ${\tilde{S}}_{2}\left(t\right)$. We implement Equation (Equation 20) as a tensor network contraction, which allows us to reach system sizes N≤1024 for any on-site Hilbert space dimension *d*. In Figure 1 (right) we can see that the difference ${\tilde{S}}_{2}^{H}-{\tilde{S}}_{2}\left(t\right)$ decreases exponentially in time (circuit depth) *t* proportionally to ${\left(2K_{d}\right)}^{t-1}$ or at a slightly quicker pace at the smallest considered system sizes N=8,16. The prefactor of the exponential decay increases monotonically with system size *N*. The growth of the prefactor becomes clearly linear beyond N=64. At N>64, the behavior of ${\tilde{S}}_{2}^{H}-{\tilde{S}}_{2}\left(t\right)$ at longer times is captured with good accuracy by Equation (Equation 28) with $\alpha_{2}=0.291\left(5\right)$.

Performing tensor network calculations for the on-site Hilbert space dimension d=3,…,13 and for system sizes up to N=256, we verify that ${\tilde{S}}_{2}^{H}-{\tilde{S}}_{2}\left(t\right)$ decreases exponentially in time, matching the prediction in Equation (Equation 28) and reproducing the analytically found value of the constant $K_{d}$. We also find that the prefactor $\alpha_{d}$ increases with *d*, as shown in the inset in Figure 1 (right). In particular, the numerical results indicate that $\alpha_{d}\overset{d\to \infty }{⟶}1/2$, in accordance with the analytical prediction for the large on-site Hilbert space dimension limit.

The results discussed in this section demonstrate that the Hilbert space delocalization under the brick wall random quantum circuits is an abrupt process. Indeed, for a given tolerance error ε≪1, the participation entropy reaches ${\tilde{S}}_{2}\left(t\right)={\tilde{S}}_{2}^{H}-\varepsilon$ at time $t_{\mathrm{HSD}}∼\ln\left(N\right)$, which scales logarithmically with the system size *N*.

## 6. Numerical Results for Any Replica

For generic values of the Rényi index *q*, the analysis presented above cannot be trivially extended. First, the calculation of ${\bar{I}}_{q}$ requires all domain wall configurations $|Q_{0}(\left\{\sigma_{j}\right\})⟫$ with $\sigma_{j}\in S_{q}$. Thus, the calculation of the IPR involves a sum over many more initial configurations than just the configurations appearing in the computation of the purities $P_{q}$, which includes only the identity permutation I and the cyclic permutation S. Moreover, $W_{2}^{\left(q\right)}$ involves contributions with negative signs and additional weights, which affect the value of $K_{d}$, resulting in more complex analysis.

Nonetheless, the main features of the participation entropy growth under brick wall quantum circuits discussed in Section 5 may be expected to hold for a generic replica number *q*. This leads us to the following conjecture:(29)$${\tilde{S}}_{q}^{H}-{\tilde{S}}_{q}\left(t\right)≃\alpha_{d,q}N\beta_{d,q}^{t}\;\;{\tilde{S}}_{q}^{H}-{\tilde{S}}_{q}\left(t\right)=\varepsilon \;\mathrm{at}\;t_{\mathrm{HSD}}∼\ln\left(N\right)\;,$$
where $\alpha_{d,q}$ is a coefficient such that $\alpha_{d,2}=\alpha_{d}$ in Equation (Equation 28), $\beta_{d,q}<1$ is a constant with $\beta_{d,2}=2K_{d}$, and ε≪1 is a fixed tolerance. The heuristic idea behind the above conjecture is that the limit d≫1 leads to positive weights in $W_{2}^{\left(q\right)}$, and, up to corrections in O(1/d), the main argument leading to Equation (Equation 28) should apply for Equation (Equation 29). We corroborate this hypothesis by analyzing the dynamics for various values of *q*. The results are summarized in Figure 2.

For qubits (d=2), we first benchmark the self-averaging properties of the circuit, as shown in Figure 2a for q=3. The self-averaging holds similarly to q=2, i.e., the annealed ${\tilde{S}}_{3}$ and the quenched average ${\bar{S}}_{3}$ of participation entropy rapidly approach each other with the increase in the circuit’s depth. We find analogous results for q=1,4 and expect that a similar phenomenology will arise for any Rényi index *q*. We therefore limit our discussion to the difference between the saturation value and the annealed average of the participation entropy, i.e., ${\tilde{S}}_{q}^{H}-{\tilde{S}}_{q}\left(t\right)$.

In panels (b–d) of Figure 2, we demonstrate that the exponential decay of ${\tilde{S}}_{q}^{H}-{\tilde{S}}_{q}\left(t\right)$ with the circuit depth *t* is observed for all considered values of the Rényi index *q*. For d=2 and q=3 (see Figure 2b), our tensor network implementation allows us to reach N≤512. Surprisingly, we find that the exponential decay of ${\tilde{S}}_{q}^{H}-{\tilde{S}}_{q}\left(t\right)$ is very accurately fitted with $\alpha_{2,3}N{\left(2K_{2}\right)}^{t}$, even though we expected a decay $\beta_{2,3}^{t}$ with a different coefficient $\beta_{2,3}$ fixed by the weights in ${|\tilde{\sigma }⟫}_{i,i+1}$, cf. Equation (Equation 29) and Section 4.

For larger values of q≥4, as well as for the von Neumann entropy limit q=1, we are limited to the relatively small system sizes N≤24 accessible with the exact numerical simulation of the brick wall random circuits. The time evolution of ${\tilde{S}}_{q}^{H}-{\tilde{S}}_{q}\left(t\right)$ for q=1,4 approaches, with increasing system size *N*, the exponential decay ${\left(2K_{2}\right)}^{t}$. The separation of ${\tilde{S}}_{q}^{H}-{\tilde{S}}_{q}\left(t\right)$ at a fixed time for different system sizes *N* suggests that the prefactor in front of the exponential decay ${\left(2K_{2}\right)}^{t}$ may be scaling proportionally to the system size *N*. However, the range of available system sizes does not allow us to fully confirm this observation.

Finally, performing exact numerical calculations for d=3 (up to N=14) and d=4 (up to N=10), we verify that the decay of ${\tilde{S}}_{q}^{H}-{\tilde{S}}_{q}\left(t\right)$ is accurately fitted with ${\left(2K_{d}\right)}^{t}$ for q=1 and q=4. This finding is analogous to the results for qubits (d=2) and corroborates further the observation that the base $2K_{d}$ characterizing the exponential decay of ${\tilde{S}}_{q}^{H}-{\tilde{S}}_{q}\left(t\right)$ is independent of the Rényi index *q*.

Overall, the numerical results for the Rényi index q≠2 presented in this section support our analysis, confirming that the approach of the participation entropy ${\tilde{S}}_{q}\left(t\right)$ to its long-time saturation value ${\tilde{S}}_{q}^{H}$ occurs exponentially in time and with a prefactor that scales extensively with the number of qubits. Hence, also for q≠2, the approach, up to a fixed tolerance, of the participation entropy to its saturation value ε occurs at time $t_{\mathrm{HSD}}∼\ln\left(N\right)$, which scales logarithmically with the system size *N*.

## 7. Discussion and Conclusions

We have investigated how the dynamics of random unitary circuits delocalize an initially localized state over a basis of the many-body Hilbert space. We combined analytical and numerical methods to calculate the time evolution of the participation entropies (Equation 1) that characterize the spread of state ρ=|Ψ〉〈Ψ| over a basis B of many-qudit systems under the dynamics of brick wall random circuits. Our main finding is that the process of Hilbert space delocalization occurs abruptly so that the long-time saturation value of participation entropies is approached up to a fixed tolerance at times $t_{\mathrm{HSD}}∼\ln\left(N\right)$, scaling logarithmically with the system size *N*. These results may appear surprising from the point of view of the close relations between the participation entropy and the entanglement entropy [26,36,43,85]. While our analytical considerations show close parallels between entanglement entropy and participation entropy calculation, the boundary conditions of the relevant statistical mechanics model are different. This difference amounts to the uncovered contrasts between the time evolution of participation entropy and entanglement entropy. Indeed, given a connected subsystem, the entanglement entropy under local circuits grows linearly in time [49], saturating at times scaling proportionally to the system size *N*. However, when the subsystem is the unions of multiple disjoint intervals, the saturation time is much faster, of the order of the biggest domain configuration. These typical configurations are the leading contribution for the two-replica IPR computation and explain the logarithmic behavior of the saturation time in *N*.

We focused on times that were short compared to the system size. A complementary path considering random circuits in the limit of t→∞ and N→∞ and demonstrating the onset of universal features in the distributions of the overlaps between the output states generalizing the Porter–Thomas distribution was considered in Ref. [86]. We note that the IPR $I_{2}$ coincides with a *collision probability* which characterizes anticoncentration properties of many-body wave functions. The anticoncentration constitutes a necessary ingredient in formal arguments of the classical hardness of the sampling problems [87,88,89,90], and analytical bounds for the collision probability under random quantum circuit evolution were previously obtained in Refs. [91,92].

Random quantum circuits may be perceived as minimal models of local unitary dynamics. Hence, similar to the entanglement entropy case [93], we expect that our results about Hilbert space delocalization extend to generic chaotic non-integrable many-body systems. Moreover, our results may provide a relevant reference point for understanding the Hilbert space delocalization in non-ergodic many-body systems, cf. [94,95,96]. The statistical invariance of the considered circuits under a product of arbitrary on-site unitary transformations shows that our results about the growth of participation entropies hold for any basis $B_{U}$ obtained from the computational basis B by a product of independent on-site rotations.

The participation entropies considered here are not only directly available in numerical simulations of many-body systems but are also of experimental relevance. Indeed, recent progress in the stochastic sampling of many-body wave functions in ultracold atomic and solid-state experiments [97,98] allows, in principle, for their direct experimental evaluation. Similarly, the quantum processors realizing quantum circuits enable high-frequency sampling of the output state [99,100], which is a basis of the cross-entropy benchmarking [101], and could enable direct measuring of the participation entropies. Finally, the process of the estimation of the IPR, which, in general, requires resources scaling exponentially with the system size *N*, may be simplified by use of appropriate quantum algorithms [102].

This work has identified the leading terms relevant to the participation of entropy growth in random unitary dynamics. There are several interesting directions for further research. Analogously to the case of entanglement entropies [49,66], one may ask a question about fluctuations of the participation entropies around the identified mean values. Properties of the dynamics of participation entropies under higher dimensional circuits are another unresolved problem. Moreover, stabilizer Rényi entropies [103,104,105,106,107], which determine the number of beyond-classical (non-Clifford) operations needed to perform a quantum task, are related to participation entropies [83]. The long-time limit of stabilizer Rényi entropies under local unitary dynamics has been recently understood [108]. However, understanding the properties of the growth of stabilizer Rényi entropies is an exciting direction for further research facilitated by the results of this work.

## Figures and Tables

**Figure 1 entropy-26-00471-f001:**
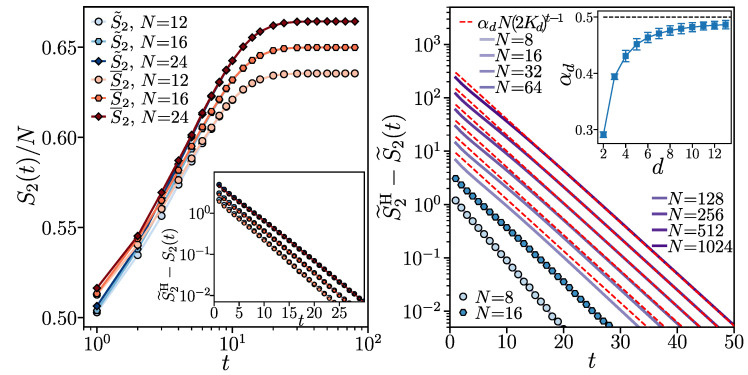
Participation entropy for the Rényi index q=2. The **left** panel shows the quenched ${\bar{S}}_{2}$ (in red) and the annealed $\tilde{S_{2}}$ (in blue) average of the participation entropy rescaled by the system size *N* and plotted as a function of circuit depth *t*. The inset shows the exponential in time decay of the difference between the average participation entropy ${\tilde{S}}_{2}^{H}$ for random Haar states and the averages ${\bar{S}}_{2},{\tilde{S}}_{2}$ as a function of time. The **right** panel shows the approach to ${\tilde{S}}_{2}^{H}$ of the annealed average $\tilde{S_{2}}$ (blue symbols) and the exact solution obtained with the tensor network approach (solid purple lines) compared with the asymptotic formula ${\tilde{S}}_{2}^{H}-{\tilde{S}}_{2}\left(t\right)=\alpha_{d}N{\left(2K_{d}\right)}^{t-1}$ (red dashed line), where, for qubits, d=2, $\alpha_{2}\approx 0.291\left(5\right)$ and $K_{2}=\frac{2}{5}$. The inset shows the prefactor $\alpha_{d}$ as function of the on-site Hilbert space dimension *d* and the $\alpha_{d}$ extrapolated from tensor network simulations, while $\alpha \overset{d\to \infty }{⟶}\frac{1}{2}$ is shown by the black dashed line.

**Figure 2 entropy-26-00471-f002:**
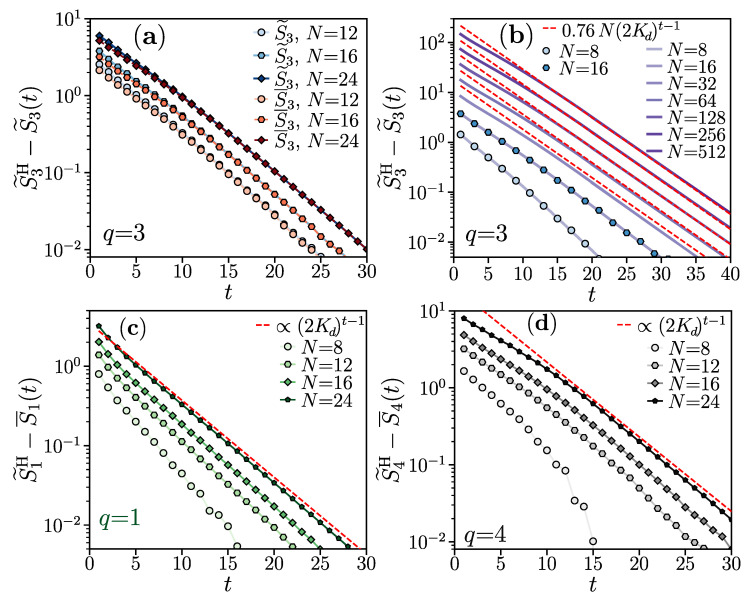
Participation entropy growth for various Rényi indices *q*. Panel (**a**) shows the approach of the quenched ${\bar{S}}_{3}$ (in red) and annealed $\tilde{S_{3}}$ (in blue) average of the participation entropy to the random Haar state value ${\tilde{S}}_{3}^{H}$ as a function of the circuit depth *t*. Panel (**b**) presents the approach to ${\tilde{S}}_{3}^{H}$ of the annealed average $\tilde{S_{3}}$ (blue symbols) calculated with the exact numerical simulation compared with the tensor network contraction results (purple lines) and the fitted formula ${\tilde{S}}_{3}^{H}-{\tilde{S}}_{3}\left(t\right)\approx 0.76N{\left(2K_{d}\right)}^{t-1}$ (red dashed line) consistent with Equation (Equation 29). Panels (**c**,**d**) show the approach of the circuit-averaged participation entropy ${\bar{S}}_{q}$ to the random Haar state value ${\tilde{S}}_{q}^{H}$, respectively, for q=1 and q=4.

## Data Availability

The code for the numerical simulations and the data are available at [77].

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
