# Peer review of "Hilbert Space Delocalization under Random Unitary Circuits"

_entropy, 2024, doi:10.3390/e26060471_

Round 1

Reviewer 1 Report

Comments and Suggestions for Authors

This work provides a comprehensive study of the IPR in Haar brickwork random circuits using a combination of analytical derivation and large-scale tensor-network contraction. The analytical derivation is based on mapping the q replica of the random circuits to a statistical model, and the tensor network is based on evolving the matrix product operator. Together, these results demonstrate that the IPA approaches its steady Haar value in a time that scales logarithmically with the system size.

The paper is well-written, and the results are presented clearly. I suggest that the authors discuss more about the relationship between the entanglement entropy and the IPR, which is governed by the same circuit, except for the boundary condition. The Rényi entropy saturates in a time that scales linearly with the system size, while the authors show that the IPR saturates in a time that scales logarithmically with the system size. This indicates that the IPA saturates well before entanglement saturates. Is this correct? How can we understand the difference? After this question is addressed, I will be happy to recommend publication.

Author Response

Please find attached the response in the PDF document.

Reviewer 2 Report

Comments and Suggestions for Authors

This paper studies how the inverse participation ratios (IPR) and the participation entropies with a value of q (Eq.(1)) behave in a random circuit (Haar random gate).(I am not sure that whether $I_q$ is the inverse participation ratios in usual sense.)

In particular, they found that the participation entropy Sq saturates exponentially quickly in time. The explicit calculation provides some information in the related research field.

The numerical calculations also seem reasonable.

 In the text, they consider q replica system to express the q-th power of probability $p_n^q$ in (Eq.(1)). Each system consists of N qudits of d states, and the states are expressed by |n>=|b^1,\cdots b^n>.

Applying a Haar distributed unitary operation $U$, the state evolves along the random circuit.

 However, the descriptions have some difficulties.

 2 line above Eq.(3): $U(d)$ denotes the group for a single qudit, but $U(d)$ above Eq.(7) should be a two spin operation (?).

 Eq.(9) must be explained.

What kind of interaction between replicas must be explained.

If we consider all the replicas are really independent, Eq. (9) is nonzero only in the same system k=k. (In Eq. (9), what is k is not explained.)

 What is concrete form of the unitary operation in the calculation is not clear, although the general action is given by (6).

The relation between Eq.(7) and Eq.(12) is not clear. In Appendix, (A1) is $U$ dependent while (A3) does not. This part must be explained in detail.

 The difference between annealed average and quench average is important, and this part must be explained in detail.

Is the annealed average means average over $I_q$? The average for one layer gives an operation which contains all the pairs with equal weights?

W_N^(q) in Eq.(8) is an averaged operation.

Is the quenched average means the average over ln[I_1] for given samples, or the limit q=1 ? $U$ contains some of pairs randomly?

 Also, in section 4, the definition of explicit definition of the annealed and quench average participation entropy (4 line above Eq.(16)) are not shown.

 They wrote that In particular, in the scaling limit N → ∞, the annealed and the quenched averages coincide, but it seems for one step, and more arguments seem necessary in the cases of more than one step.

 In Eq.(16), $U_{ij}$ is still general two-body gate, and for concrete calculation an explicit form is necessary. This part is not clear.

 In Eq.(19), Haar average of t layers is expressed by unit of Haar average of one unit (Rq.(8)).

Average of a production of operation may not be equal to production of averaged operation (?).

In this case, do the anneal and quench averages also give the same results?

 In the case that the annealed and the quenched averages coincide, the process is essentially in a uniform averaged layer, and effect of randomness is rather irrelevant, and only operations all over spins is essential (?)

 Definition of $|\tilde{\sigma}>>_{i,i+1}$ is not given, because

allocations of U_{ij} is not explained well (brick means one dimensional alternate arrangements(?)).

 Definition of ${\cal W}_t$ at the line 146 is not given.

 2 lines below Eq.(26), they consider the case d>>N. Is it reasonable?

Later they studied the case N>>1 for a fixed d numerically, which also gives exponential decay of the difference. Thus, the exponential decay was confirmed. But does the argument for the case d>>N helps to understand the results?

 In numerical calculation, it is preferable to demonstrate the behavior $S_2$ in many samples of randomly fixed arrangements, which is real quench process.

 These points should be explained reasonably before the publication.

Author Response

Please find the response attached. 

Reviewer 3 Report

Comments and Suggestions for Authors

In the paper the authors calculate the dynamics of an average, so-called, inverse participation ratio, which together with the second figure of merit studied here, the participation entropy, measure delocalization of a quantum state in a chosen basis. The average is taken over the Haar measure random unitary transformations of a chosen structure, including the so-called brick-wall quantum circuits, in which the local unitary transformation in a single layer act on neighboring qudits (elementary quantum systems) ordered in a chain starting either from even or from odd qudit. The main theoretical driver that allows for analytical formula derivation is the theory of Weingarten, ref. 79. The theory is supported and completed in theoretically difficult cases by numerics based on tensor network formalism.

The most significant result is recognizing that the speed of delocalization measured by the studied quantities is exponentially quick in time, which is encapsulated in formula 2, justified analytically or numerically in the paper. This dynamics can be surprising as the entanglement entropy which is a quantity somehow like the participation entropy does not behave like that.

The authors conclude their report discussing some applications. Although at the first glance the random unitary dynamics especially given by the brick-wall circuit may seem artificial and far from natural dynamics which we find in physics, the authors indicate interesting applications including sampling from distributions which could be hard to achieve classically at the same speed. Sampling from exotic distribution is indeed an interesting topic with current world applications.

The paper is well structured and written. It is clear and, up to my best experience, mathematically rigorous and correct. In my opinion it satisfies scientific standards. Therefore, I recommend publishing it. I did not find any flaws and for me the paper can be published as it is. 

Author Response

We thank the Referee for their careful reading and positive recommendation of our work. 

Round 2

Reviewer 2 Report

Comments and Suggestions for Authors

The paper has been revised. But still some parts are not clear.

However, the paper contains various technical interesting points and I agree with to  publish the paper.  

My question on the explicit form of $U$, namely the operation of $U_{i,j}^{k,l}$ in Eq. (19) is not yet given. “random Haar gates” means taking a unitary operation by the Haar measure, but it does not specify what the unitary operation. Also, where “uniform” means $i$ and $j$ distribute uniformly in the lattice or $ij$ pairs distribute on a given lattice is not stated.

For my question “In numerical calculation, it is preferable to demonstrate the behavior $S_2$ in many samples of randomly fixed arrangements, which is real quench process.”

they wrote

“If we understand this remark correctly, this is exactly what we did in Fig. 1 Left, and Fig. 2(a).”.

I did not realize this point. In this case, they should have explained this part more explicitly, e.g., write how many random samples they calculated to quench average and how to construct each sample, etc..

Final general question to be concerned is the following.

In the case where the quench process and anneal process give the same process, effects of the randomness do not work seriously. Whether this is a special property of Haar measure is an interesting problem.

Author Response

We have performed a minor revision of the manuscript to resolve the remaining problems indicated by the Referee. Specifically:

1. The indices specifying qudits on which two-body gate defined in Eq. (17) is acting could be confused with the indices of a generic two-body gate in Eq. (19). To solve this issue, we changed the indices denoting the qudits on which the gate is acting to $m,n \in \{1,...,N}$. At the same time, Eq. (19) involves the indices $i,j,k,l \in \{0,1,..., d-1\}$, which denote the states of the qudits.

2. The structure of the brick-wall cicruit is defined in Eq. (17). The sentence just after Eq. (17) explicitly states that $U_{m,n}$ is a randomly choosen 2-body gate acting on the fixed qubits $m$ and $n$. The spatial structure of the brick-wall circuits is defined in Eq. (17). Hence, it should be clear that the brick-wall circuit involves two body gates acting on neighboring qudits. Moreover, since each of the two-body gates $U_{m,n}$ is drawn randomly with the Haar measure from unitary group, it has no fixed form that we could provide explicitly.

3. Regarding Fig. 1 Left and Fig. 2(a). We have explicitly stated that the number of considered circuit realizations is $\mathcal{N}_H=10^4$. Additionally, the description around Eq. (17) clearly specifies how each realization of the circuit is generated.

4. We agree with the Referee that the concidence of the quenched and annealed averages is an interesting feature of the random Haar circuits. Since the local random Haar circuits provide minimal models of chaotic many-body dynamics, we conjecture that this property may carry over to generic thermalizing many-body systems. At the same time, there may exist settings in which that it is not the case, for instance in quantum systems in which the ergodicity is broken. We leave these questions for future investigations.

We hope that the present version of our manuscript is suitable for publication in Entropy.